# FRAPPE: FAST RAG-INSPIRED PROMPT EVAPORATOR

## ABSTRACT

Large Language Models (LLMs) have demonstrated remarkable performance in various tasks such as multi-document QA, summarization, and text classification. This has been achieved in part by recent advancements in prompt engineering and in-context learning (ICL), enabling LLMs to consume tens of thousands of input tokens as the supported context for the given query. However, this creates higher computational costs, longer latency, and potential performance degradation. To address these issues, we propose a task-agnostic and efficient approach called "Fast RAG Inspired Prompt Evaporator", or FRAPPE, to significantly reduce LLMs' latency, memory requirement, and computation by compressing input tokens. Unlike many other proposed approaches for prompt compression, our method does not rely on any large model for computing conditional probabilities, and data preparation is fast with negligible memory requirements. In particular, our approach first pre-processes the input data, categorizes and ranks phrases based on their informativeness, and finally selects the highest-ranked phrases to generate highly compressed and extractive input. We show the efficacy of our approach through a comprehensive set of experiments on public datasets and benchmarks. For instance, on the summarization task of the MeetingBank dataset, at a compression rate of 70%, our proposed approach achieves performance similar to the full context while performing compression up to 4 times faster than the contemporary state of the art compression algorithms. We extend FRAPPE to create the Context-Aware FRAPPE algorithm, which incorporates task-specific information when ranking phrases, which further improves performance of downstream tasks using compressed text. Additionally, we demonstrate that the use of FRAPPE can reduce toxicity by close to 50% relative to the original text by removing extraneous vitriolic phrases, in contrast to other compression methods, which often increase toxicity.

## 1 INTRODUCTION

Large Language Models (LLMs) have progressed to exhibit strong performance across various tasks such as multi-document QA, information retrieval, in-context learning and reasoning, code completion, and document summarization (Brown et al., 2020). These LLMs, which are based on the Transformers (Vaswani et al., 2017) architecture, have achieved such strong performance thanks to intelligent architectural and training choices. Moreover, recent prompting techniques, including In-context Learning (ICL) (Dong et al., 2022), Chain-of-Thought (COT) (Wei et al., 2022), and Retrieval Augmented Generation (RAG) (Lewis et al., 2020) have enabled LLMs to handle complex queries thanks to the supporting context provided by lengthy inputs with tens of thousands of tokens. However, there are inherent constraints on LLM performance, such as limited context window size[1] and the quadratic complexity of the attention mechanism. Long prompts increase the computational challenge for LLMs, resulting in longer processing times and potentially inferior LLM performance (Xiong et al., 2023). Similarly, as the window size of an LLM increases, it may be less sensitive to the related information in the query (Shi et al., 2023). These restrictions have spurred many attempts to reduce the memory and computation cost of the Transformer models using architectural

---

[1]Recent LLMs models, like Claude 3 Haiku, may allow for up to 200K tokens, but that can be limited for multi-document QA tasks.

optimization such as sparse attention (Child et al., 2019), local dense attention (Beltagy et al., 2020), grouped attention (Burchi & Vielzeuf, 2021), and Flash Attention (Dao et al., 2022).

An alternative approach to increasing LLM efficiency is reducing the input token length through *prompt compression*, which prunes irrelevant or non-informative tokens without sacrificing performance. Here, we focus on this latter viewpoint. Prompt compression can be *task-aware*, where non-informative tokens are pruned based on specific downstream tasks, often improving performance. One canonical example of this kind of approach is in question answering. However, this can be inefficient for RAG-based applications, as it requires designing multiple compression schemes per task. A more efficient option is task-agnostic compression, which removes tokens without considering the task or query. This approach leverages the redundancy in human languages(Shannon, 1951), which might not be useful for LLMs to generate texts. Proposed task-agnostic compression methods typically consider some notion of information-theoretical measures such as surprisal (self-information) or perplexity provided by a smaller language model (Li et al., 2023b; Jiang et al., 2023a). A potential problem with this measure is that it may be sub-optimal and model-dependent. Moreover, using causal LMs to compute the measures is limited to one direction in the context, which is not aligned with many tasks, requiring a full context.

This paper proposes a task-agnostic prompt compression method inspired by RAG, called "Fast RAG Inspired Prompt Evaporator", or FRAPPE. The method chunks input text into smaller phrases, removing common repetitions and fillers. Using embedding vectors, it prunes phrases aligned with low-information phrases. The remaining phrases are then ranked by saliency, with the top ones forming the compressed text.

We summarize our contribution as follows:

- Inspired by the RAG pipeline, we propose FRAPPE, a fast, modular, and task-agnostic compression algorithm up to four times faster than state-of-the-art methods.
- Unlike contemporary algorithms, FRAPPE does not use an information-theoretical measure or LLM to compute the conditional probabilities and data preparation, improving computational efficiency.
- Comprehensive experiments on public datasets for tasks like summarization, multi-document QA, conversation, in-context reasoning, and code completion show FRAPPE's superior or comparable performance against strong baselines and full context prompts.
- We demonstrate an extension of FRAPPE using task-specific context for phrase pruning.
- We present a study that shows that FRAPPE compression can *reduce* text toxicity.

## 2  RELATED WORK

The quadratic computation complexity of Transformers may result in an input length-dependent increase in the time to generate responses by LLMs, increasing computational costs. ATo address this, various methods have been developed to improve efficiency, including model compression techniques like pruning, knowledge distillation, quantization, and low-rank factorization (Zhu et al., 2023) and using optimized implementations of the attention mechanism such as sparse attention (Child et al., 2019), local dense attention (Beltagy et al., 2020), grouped attention (Burchi & Vielzeuf, 2021), and Flash Attention (Dao et al., 2022). Additionally, LLMs' limited context windows restrict the use of prompt engineering methods like Chain-of-Thought (COT) (Wei et al., 2022), and Retrieval Augmented Generation (RAG) (Lewis et al., 2020). Recently, data-centric methods have emerged, focusing on selecting diverse, informative examples for efficient learning.

In this regard, prompt compression has recently emerged as a promising data-centric method that selects the most informative documents, phrases, words, or tokens using a coarse to granular pruning strategy in task-aware or task-agnostic ways. The former methods are tailored to a specific downstream task, usually resulting in improved performance. However, they usually require the design of multiple compression schemes for every single task, which may increase the complexity of algorithm deployment. On the other hand, task-agnostic compression methods remove tokens without considering the query and/or downstream task, and they are more generalizable to multiple tasks. However, they can yield sub-optimal performance and model-dependent results. Moreover, using causal LMs to compute the probabilistic measures is limited to one direction in the context, which

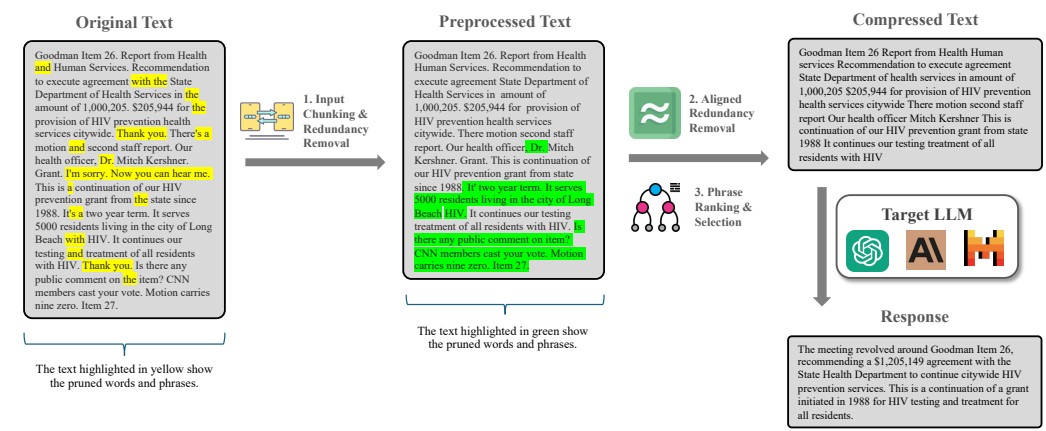

Figure 1: A high-level overview of our multi-step and modular approach with an example.

is not aligned with many tasks requiring a full context. A key prompt compression approach is token pruning (Kim et al., 2022; Li et al., 2023a), where a language model is trained to compress prompts into a smaller number of tokens. Other methods include soft prompt compression and context pruning methods (Wingate et al., 2022; Mu et al., 2024; Chevalier et al., 2023; Ge et al., 2023; Anagnostidis et al., 2024). These methods are based on training a small set of weights introduced by soft-prompt or summary vectors. Other approaches are based on information-theoretical measures, relying on LLMs to compute the conditional probabilities used by these metrics, and are model dependent (Li et al., 2023b; Jiang et al., 2023a; Pan et al., 2024). In particular, (Pan et al., 2024) use a data distillation approach generated by GPT-4 to train their token classifier. Since GPT-4 can struggle to preserve key information (Jiang et al., 2023b) and sometimes modifies content, they proposed a data-controlling mechanism to reduce these effects. Lastly, RL-based approaches (Huang et al., 2023; Jung & Kim, 2023) use a reward model to find an optimal policy to remove or retain a token in the input prompt. Compared to the above approaches for prompt compression, FRAPPE does not rely on any LLM to generate content, and it also does not require any LLM to compute the conditional probabilities or a reward model to select or prune a token. Thus, it is unbiased regarding the hallucinated content, fast, highly efficient, and cost-efficient, as shown in the experimental section, making it a desirable and simple choice for deployment in production scenarios.

## 3 PROPOSED APPROACH

This section details FRAPPE which treats a prompt as a tuple of instructions, context (aka demonstration), and queries. We use $\overline{\tau} = 1 - \tau$ to denote the target compression rate we want to achieve (fraction of pruning tokens), and $\tau$ to denotes the fraction of remaining tokens. Our method is inspired by the RAG pipeline, in which context is first chunked, indexed, and stored using an embedding model, and non-informative phrases are pruned by comparing them against a database of redundant words and phrases. As illustrated in Figure 1, our approach has two phases: Pre-processing and Compression. In the Pre-processing phase, the input prompt is chunked into smaller pieces (i.e., phrases), and those ones found in the pre-defined database of phrases, which we call *redundancies*, are removed from each piece. The Compression phase itself has two components. First, all phrases closely aligned with the redundancies are identified and deleted using a similarity approach. Second, the remaining phrases in the input are ranked by their saliency, and the top-ranked phrases are selected according to the rate $\tau$. Thus, the input prompt has been compressed in an extractive way. The compressed input can now be sent to a target LLM for accomplishing the desired task[2]. We now provide more details of each step in our approach.

---

[2]If the downstream task is summarization, we can use the compressed input as the extractive summary if the token count is sufficiently small.

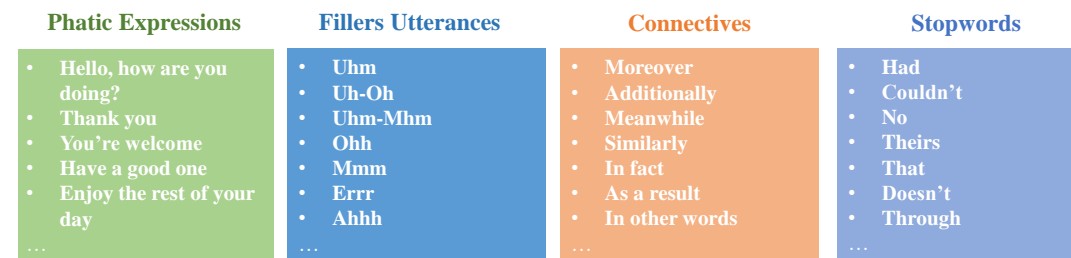

Figure 2: Four groups of redundancies used for cleaning the input prompt.

## 3.1 INPUT CHUNKING AND REDUNDANCY REMOVAL

The input prompt is segmented into smaller chunks using five punctuation marks: ".", ",", ";", "!", "?". These chunks, or *phrases* can be sentences or other expressions comprising of one or more words, resulting in $N$ number of tokens. We then remove articles (e.g., a, an, the) from all phrases[3]. Next, repeated phrases and those included in one of our defined *redundancy groups* are cleaned out. Figure 2 shows four defined redundancy groups with some examples. This step is crucial for compressing transcribed speech, conversations, or meeting transcripts.The redundancy groups— Phatic Expressions, Filler-Utterances, Connectives, and Stop Words were created by prompting GPT-4 and curated through several iterations for a comprehensive collection. Now, we provide more details about creating each redundancy group (please section A.7 for more details:

**Phatic Expressions.** These phrases facilitate social interaction rather than conveying information. We started by asking GPT-4 for a list of Phatic Expressions and expanded it with variations and examples for greetings and conversation closures. After careful curation, we compiled around 90 expressions, diversifying the list with examples from online resources.

**Filler Utterances.** The second category involved asking GPT-4 for a comprehensive list of Filler Utterances in English, targeting redundancies in casual conversations. We compiled about 30 words/phrases like "huh", "mmm", "uhm", and "ah".

**Connectives.** We prompted GPT-4 to generate a list of connective words, conjunctions, and transitional phrases in English. This produced an extensive list, highlighting groups like Comparative (e.g., "similarly"), Additive (e.g., "and"), Contrastive (e.g., "but"), and others including Conditional, Summarize, Illustrative, and Time categories.

**Stopwords.** We have utilized the NLTK Library (Bird et al., 2009). for this category.

After removing all phrases in the redundancy groups, we end up with $N_c$ tokens pruned from $N$ initial tokens; hence, the compression rate by the end of this stage is given by $\overline{\tau_c} = \frac{N_c}{N}$.

## 3.2 ALIGNED REDUNDANCY REMOVAL

Each redundancy group contains phrases that are variations conveying similar information. For instance, in "Phatic Expressions," phrases like "How are you doing?" and "How you doing?" are semantically alike. Since it's impractical to list all variations, we consider a subset of common examples in each category (some of which are shown in Figure 2). This forms the redundancy database. We then use an embedding model to extract the embedding vectors of all phrases remaining after the prior stage. We can compute the cosine similarity of the remaining phrases to those phrases included in our redundancy database[4]. Phrases too similar to redundant ones are pruned at this stage, resulting in a clean input prompt.

This approach is similar to the RAG retrieval mechanism for finding related contexts to a query. Using phrase embeddings is cheaper and faster than token-level embeddings because there are fewer phrases than tokens, as noted by (Li et al., 2023b). Moreover, we observe that eliminating

---

[3]In section A.5, we show that removing articles and punctuation can simply compress the input text by 16%

[4]Any other type of similarity measure can be used. However, cosine similarity is by far the most popular one in the context of embedding models as it is a scale-invariant measure.

phrases does not impact the fluency of the text as much as pruning individual tokens does. After this phase, $N_r$ tokens will be removed from $N - N_c$ remaining tokens from the previous stage, so we have $\overline{\tau_r} = \frac{N_r}{N - N_c}$. In our approach, we present the main results by the zero-shot Sentence-Transformers (Reimers & Gurevych, 2019) or SBERT model (all-MiniLM-L6-v2), which is efficient in speed and memory. In the ablation study, we show the effect of other embedding models such as General Text Embeddings (GTE)-large model (Li et al., 2023c), one of the leading embedding models in MTEB leaderboard[5].

### 3.3 PHRASE RANKING AND SELECTION

At the final stage, we compress the remaining phrases by ranking and selecting the most informative ones, resulting in $N_e$ pruned tokens, where $\overline{\tau_e} = \frac{N_e}{N - N_c - N_r}$. Putting all together, the total number of pruned tokens at the end will be given by $\tau N = N_c + N_r + N_e$. Here, we have used a ranking algorithm to sort the phrases in terms of their *importance or informativeness* in the input. Since our main goal is to compress the input tokens with a low time complexity while producing high-quality results, we focus on fast and efficient graph-based ranking algorithms. In particular, the phrases are represented as a set of vertices $V$ in a weighted graph $G = (V, E)$, where $E$ denotes the set of edges, i.e., Edge $e_{ij}$ from node $V_i$ to node $V_j$ is weighted by the similarity score (cosine similarity between embedding vectors of phrases) from the previous stage. Thus, the entries of the adjacency matrix or similarity matrix (**SM**) is given by $\mathbf{SM}_{ij} = w_{ij} = \text{CosSim}(V_i, V_j)$. Having constructed the similarity matrix, we can now rank the nodes in the graph and select the top ones. To calculate nodes' (or phrases') salience, we use the concept of *Node Centrality*.

Different methods have been proposed for calculating the centrality of nodes. This includes algorithms such as TextRank or LexRanks algorithms (Mihalcea & Tarau, 2004; Erkan & Radev, 2004) (which are adapted from PageRank (Brin & Page, 1998)), and more recently, PACSUM (Zheng & Lapata, 2019), FAR (Liang et al., 2021), STAS (Xu et al., 2020), and HipoRank (Dong et al., 2021). In the TexRank-type of algorithms, an undirected graph G is considered, and the importance score of nodes is iteratively updated based on the combination of the current importance scores and the values of edges until no significant changes are observed. This can be seen as finding the stationary distribution of the Markov chain where the transition matrix is defined based on the similarity matrix and a damping factor to ensure the underlying graph is irreducible. Hence, the nodes' Centrality is computed by finding the leading left eigenvector (corresponding to the largest eigenvalue). Experiments show that methods such as PACSUM, FAR, STAS, and HipoRank exhibit similar performance (Xu et al., 2020)[6]. These methods establish a directed underlying graph G as they account for the order of phrases in a text. As a result, *asymmetric* centrality of node $V_i$ defined as

$$\text{Centrality}(V_i) = \lambda_1 \sum_{j<i} \mathbf{SM}_{ij} + \lambda_2 \sum_{j>i} \mathbf{SM}_{ij}, \tag{1}$$

where $\lambda_1$ and $\lambda_2$ are two hyper-parameters to adjust the impact of previous and last content, and they are set such that $\lambda_1 + \lambda_2 = 1$. if $\lambda_1 = \lambda_2$, the above asymmetric node centrality becomes degree centrality (a symmetric centrality). Once the centrality of all nodes has been calculated, the top $N_e$ nodes with the largest centrality score (i.e., most salient phrases) are selected as the final compressed input prompt. Here, we have utilized (Hagberg et al., 2008) for the implementation of TextRank. However, instead of ranking individual pieces of text, we rank phrases. In the ablation study, we study the impact of using an asymmetric degree centrality method (PACSUM) as a ranking algorithm.

## 4 EXPERIMENTS

This section demonstrates our method's performance through comprehensive experiments. The embedding model is not fine-tuned to show its task-agnostic and generalizable nature to out-of-domain distributions. We briefly describe tasks, datasets, and our experiment setup and defer the details to the appendix.

---

[5]https://huggingface.co/spaces/mteb/leaderboard

[6]HipoRank has better performance (Dong et al., 2021) but operates in a hierarchical level, so it is not as fast as PACSUM.

**Tasks and Datasets** We have tested our algorithm on various tasks, including summarization, multi-document question answering (QA), in-context reasoning (here, answering math and science questions), and code completion. We use appropriate evaluation metrics corresponding to each task, including BLEU (Papineni et al., 2002), Rouge-1/2/L (Lin, 2004), METEOR (Banerjee & Lavie, 2005), and BertScore-F1 (Zhang et al., 2019) for summarization, multi-document QA, Exact Match (EM) score for in-context reasoning, and Edit Similarity (similarity of two strings based on the number of insertions, deletions, and substitutions) for code completion task.

Our experiments include the following datasets and benchmarks.

**Arxiv Preprint** (Cohan et al., 2018) is a repository consisting of Arxiv papers spanning topics such as Physics, Astrophysics, Biology, and Chemistry (we created a test set using the first 500 example articles in this dataset). **MeetingBank** (Hu et al., 2023) includes 862 meeting transcripts from six cities or municipalities in the test set. **ZeroSCROLLS Benchmark** (Shaham et al., 2023) is a benchmark for multiple tasks that require long context understanding. **GSM8K** (Cobbe et al., 2021) includes graduate math questions and corresponding answers. **BBH** (Suzgun et al., 2022) is a suite of language and symbolic reasoning tasks. **ShareGPT** (sha, 2023) includes the first 600 conversation transcripts from a dataset of human interactions with LLMs. **LongBench Benchmark** (Bai et al., 2023) is a benchmark for multitask assessment of LLMs' long context understanding capabilities.

**Target LLMs and Baseline Models.** We report the performance of our compression method with various target LLMs, including GPT-3.5 Turbo, Mistral.mixtral-8x7b-instruct-v0:1, and Claude-3 Haiku-20240307-v1:0, comparing against the state-of-the-art compression methods, including Selective-Context (Li et al., 2023b) and both small and large LLmLingua-2 models (Pan et al., 2024).

In all the following tables, the time column denotes the per-input compression time in seconds (average time over all the input data). Also, "Uncomp" means the uncompressed input, "SC" stands for Selective-Context, and "Lingua2-S/L" denote the LLMLingua-2-small/large models.

## 4.1 RESULTS

For experiments in this section, we use zero-shot SBERT[7] for the embedding model to extract the embedding of phrases, and we use the cosine similarity for the entries of the similarity matrix.

**Arxiv articles, MeetingBank transcripts, and ShareGPT conversations.** Table 1 shows summarization results for both original and compressed inputs using FRAPPE and other SOTA algorithms with GPT-3.5 Turbo. The compression rate was set to 70%. For the Arxiv articles (500 examples from the `ccdv/arxiv-summarization` test set), where abstracts serve as the ground-truth summaries, FRAPPE outperforms other algorithms and even surpasses uncompressed articles, as the LLM's limited context window struggles with full-length articles. For the MeetingBank transcripts, both LLMLingua-2 models were fine-tuned on this dataset, but this serves as an out-of-domain (OOD) experiment for both our approach and the Selective-Context method. Despite not fine-tuning our embedding model, FRAPPE demonstrates comparable performance to LLMLingua-2. Due to the lack of ground truth for ShareGPT conversations, we used the target LLM (GPT-3.5 Turbo) to summarize the conversations, treating these summaries as the ground truth. In this context, FRAPPE outperforms other compression algorithms and consistently achieves the shortest running time, aligning with our goal of developing a fast and efficient compression method. For experiments with Claude-3 HAIKU and Mistral models, see Section A.3.

**ZeroSCROLLS benchmark.** We also evaluated the ZeroSCROLLS validation set, which contains approximately 20 examples per dataset. We maintained a compression rate of 70% and used the provided instructions and queries. Table 2 shows the scores achieved using GPT-3.5 Turbo, Claude 3 Haiku, and Mistral-8x7B Instruct v0.1 as the target LLM. For both GPT-3.5 Turbo and Mistral, FRAPPE outperforms all other algorithms, including uncompressed text. For Claude 3 Haiku, our method scores within 5% of the highest (uncompressed) score. Notably, FRAPPE compresses the full validation dataset over twice as fast as the next fastest algorithm.

---

[7]https://www.sbert.net/

| Methods | Rouge-1 | Rouge-2 | Rouge-L | BERTScore | METEOR | BLEU | Time(s) |
|---|---|---|---|---|---|---|---|
| | | | **Arxiv Articles** | | | | |
| Uncomp | 0.3246 | 0.1083 | 0.1785 | 0.7096 | 0.2423 | 0.0295 | – |
| SC | 0.3142 | 0.0860 | 0.1756 | 0.7380 | 0.2141 | 0.0149 | 3.90 |
| Lingua2-S | 0.3386 | 0.0973 | 0.1879 | 0.7946 | 0.2514 | 0.0170 | 0.23 |
| Lingua2-L | 0.3274 | 0.0940 | 0.1803 | 0.7664 | 0.2441 | 0.0174 | 0.58 |
| **(FRAPPE)** | 0.3506 | 0.1120 | 0.1952 | 0.7739 | 0.2539 | 0.0287 | 0.14 |
| | | | **MeetingBank Transcripts** | | | | |
| Uncomp | 0.2830 | 0.1268 | 0.2100 | 0.8510 | 0.2935 | 0.0466 | - |
| SC | 0.2502 | 0.0693 | 0.1673 | 0.8417 | 0.2352 | 0.0110 | 2.76 |
| Lingua2-S | 0.2676 | 0.0950 | 0.1841 | 0.8474 | 0.2754 | 0.0240 | 0.11 |
| Lingua2-L | 0.2673 | 0.0947 | 0.1838 | 0.8474 | 0.2749 | 0.0238 | 0.23 |
| **(FRAPPE)** | 0.2632 | 0.1014 | 0.1902 | 0.8456 | 0.2605 | 0.0301 | 0.06 |
| | | | **ShareGPT Conversations** | | | | |
| SC | 0.3918 | 0.1975 | 0.2286 | 0.8276 | 0.2299 | 0.0198 | 0.71 |
| Lingua2-S | 0.4333 | 0.1582 | 0.2535 | 0.8277 | 0.2855 | 0.0357 | 0.06 |
| Lingua2-L | 0.4375 | 0.1855 | 0.2805 | 0.8223 | 0.2793 | 0.0388 | 0.14 |
| **(FRAPPE)** | 0.4545 | 0.2195 | 0.3212 | 0.8282 | 0.2798 | 0.0712 | 0.05 |

Table 1: Comparing FRAPPE with the SOTA methods on Arxiv articles, Meetingbank, ShareGPT using GPT-3.5 Turbo model as the target LLM and the compression rate of 0.7.

| Methods | ZeroSCROLLS Score | | | Time(s) |
|---|---|---|---|---|
| | GPT-3.5 Turbo | Claude 3 Haiku | Mistral - 8x7B | |
| Uncomp | 25.90 | 22.33 | 24.35 | – |
| SC | 26.67 | 15.69 | 22.42 | 6.00 |
| Lingua2-S | 20.08 | 22.01 | 23.32 | 0.29 |
| Lingua2-L | 18.35 | 20.98 | 23.17 | 0.67 |
| **(FRAPPE)** | 30.95 | 21.30 | 24.26 | 0.14 |

Table 2: Comparing FRAPPE with the SOTA methods on ZeroSCROLLS dataset and the compression rate of 0.7.

**GSM8K and BBH.** To further evaluate our approach on challenging reasoning tasks, we applied FRAPPE and other algorithms to the GSM8k dataset using a complex multi-step CoT prompt (Fu et al., 2022). [8] Additionally, we tested the compression algorithms on the BBH benchmark, selecting 16 tasks (please see the section A.2 in the appendix for the selected tasks). We used the Claude-3

---

[8]We did not enforce punctuation preservation across compression algorithms to ensure a fair comparison.

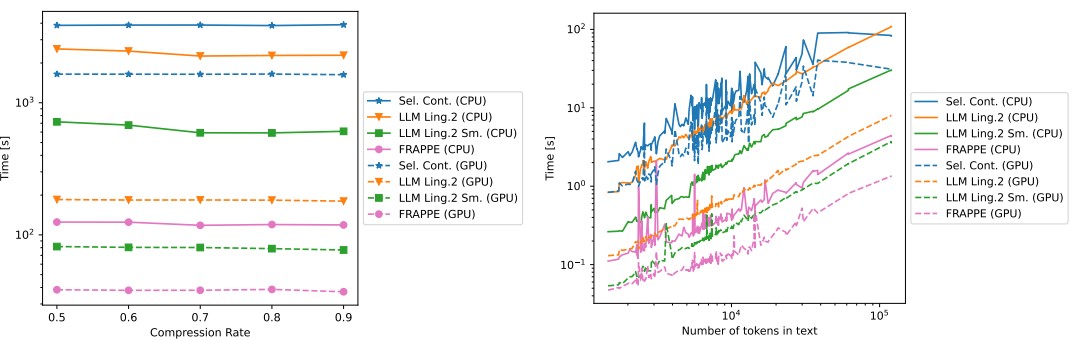

Figure 3: **Left.** Total time to compress on the ZeroSCROLLS validation set for different compression rates, algorithms, and computational hardware. **Right.** Compression time versus input length in tokens for the ZeroSCROLLS validation set with compression rate set to 70%.

| Methods | Exact Match (EM) | | | |
|---|---|---|---|---|
| | GSM8K | Time(s) | BBH | Time(s) |
| Uncomp | 0.8006 | - | 0.5132 | - |
| SC | 0.7680 | 2.33 | 0.5087 | 1.61 |
| Lingua2-S | 0.7468 | 1.18 | 0.5085 | 0.94 |
| Lingua2-L | 0.7453 | 2.49 | 0.5086 | 2.12 |
| (FRAPPE) | 0.7581 | 0.73 | 0.5103 | 0.72 |

| Methods | Size (GB) |
|---|---|
| SC | 0.511 |
| Lingua2-S | 0.710 |
| Lingua2-L | 2.236 |
| (FRAPPE) | 0.099 |

Table 3: **Left.** Comparing FRAPPE with the SOTA methods on GSM8K and BBH dataset using Claude-3 Haiku as the target LLM and the compression rate of 0.7. **Right.** Memory requirement for compression methods.

Haiku model for both datasets, reporting the EM metric for GSM8k and the average EM across all 16 tasks for BBH. As illustrated in the left panel of Table 3, our method is not only faster than others but also matches or exceeds their performance, remaining competitive with uncompressed results for GSM8k.

The right panel of Table 3 displays the memory requirements for all compression models. FRAPPE has a significantly lower GPU memory overhead, using only 99 MB—5.2x less than the second smallest method, Selective-Context.

## 4.2 COMPRESSION LATENCY EVALUATION AND MEMORY REQUIREMENTS

The experiments demonstrate that FRAPPE outperforms other algorithms in speed. However, compression time may depend on factors such as compression rate, text length, and hardware (CPU/GPU). We investigate these elements using the ZeroSCROLLS validation set, which covers a wide range of text lengths. This dataset spans about two orders of magnitude of text lengths. The left plot in Figure 3 shows the total time taken to compress our test set for 5 different compression rates on both a CPU-only machine and on a machine with an NVIDIA A100 GPU (Choquette et al., 2021). While the compression rate has little impact on the four algorithms, FRAPPE's speed is notable. Interestingly, the only algorithms that are faster than FRAPPE on a CPU are FRAPPE on a GPU and LLMLingua-2 Small on a GPU. The right plot in Figure 3 shows the dependence of compression time on text length. Each line in this figure represents 269 distinct length-time measurements (one for each example in the dataset) for the given algorithm and environment. First, one can note the roughly linear dependence of compression time on text length, which is true for all algorithms and both CPU-only and GPU-enabled configurations.

Another aspect shown in this plot is the stability of the algorithms. The LLMLingua-2 algorithms show consistent lines with minimal deviations from their linear dependence on text length. Selective Context exhibits more variability in timing for texts of similar lengths. FRAPPE is mostly stable, with only a few instances of slow convergence due to the power iteration algorithm used for computing the leading eigenvector of the similarity matrix. Despite these occasional slowdowns, FRAPPE remains the fastest algorithm on average.

## 4.3 CONTEXT-AWARE FRAPPE

Frappe is effective for summarization but can miss query-specific details due to its centrality-based compression. This oversight is due to the diverse nature of queries, which may not always align with the central theme of a document.

Context-Aware (shown as Cont-Aware in the following tables) FRAPPE addresses this issue by including the query prompt with the input text, ensuring phrases similar to the query are retained during redundancy removal. The algorithm calculates a query similarity vector between the query and phrases, using it as initial weights in a Personalized PageRank algorithm. Unlike Frappe, which solely focuses on centrality, Context-Aware FRAPPE generates a new centrality score that considers the relevance to the specific query. We evaluate the performance of the Context-Aware FRAPPE algorithm with Uncompressed text, Frappe, and the LLMLingua2-small model, using the GPT 3.5-Turbo Language Model, with a 70% compression rate for all algorithms.

Two evaluation methods were used: the Full Document approach, where the entire document is processed by the LLM, and the Retrieval-Augmented Generation (RAG) approach. In RAG, the document is divided into passages and only the most relevant ones to the query, under the context length of GPT 3.5 Turbo (4096 tokens), are processed. The evaluation was conducted on three different datasets: 2wikimqa, hotpotqa, and musique with The F1 score, a balanced measure of precision and recall, used as the evaluation metric.

| Methods | Full Document | | | RAG implementation | | |
|---|---|---|---|---|---|---|
| | 2wikimqa | hotpotqa | musique | 2wikimqa | hotpotqa | musique |
| Uncomp | 0.365 | 0.445 | 0.209 | **0.427** | 0.492 | 0.229 |
| Lingua2-S | 0.364 | 0.505 | 0.23 | 0.365 | 0.51 | **0.304** |
| **FRAPPE** | 0.354 | 0.484 | 0.235 | 0.342 | 0.472 | 0.239 |
| **Cont-Aware FRAPPE** | **0.382** | **0.507** | **0.284** | 0.406 | **0.512** | 0.272 |

Table 4: This table compares the Uncompressed documents with three compression algorithms: FRAPPE, Context-Aware Frappe, and Lingua2-S. All methods have a compression rate of 0.7. The comparison includes two approaches: Full documents and a RAG approach, using three datasets (2wikimqa, hotpotqa, musique). The evaluations use GPT 3.5-Turbo as the downstream LLM.

Additionally, we employed the Claude3 Haiku model with a 200,000-token context, allowing full document processing without segmentation. Context-Aware Frappe demonstrated a performance similar to processing the full text, but with a significant cost reduction of approximately 3x with a 70% compression rate, offering economical efficiency without quality loss. Though slightly slower than FRAPPE due to query-specific computations, Context-Aware Frappe is still about twice as fast as the LLMLingua2-small model.

| Methods | 2wikimqa | | hotpotqa | | musique | |
|---|---|---|---|---|---|---|
| | F1 | Time(s) | F1 | Time(s) | F1 | Time(s) |
| Uncomp | 0.5 | - | 0.514 | - | 0.313 | - |
| Lingua2-S | 0.498 | 0.184 | 0.5 | 0.338 | 0.234 | 0.413 |
| **FRAPPE** | 0.443 | 0.095 | 0.489 | 0.145 | 0.268 | 0.168 |
| **Cont-Aware FRAPPE** | 0.505 | 0.104 | 0.529 | 0.149 | 0.285 | 0.177 |

Table 5: This table compares the Uncompressed documents with three compression algorithms: FRAPPE, Context-Aware Frappe, and Lingua2-S. All methods have a compression rate of 0.7 using three datasets (2wikimqa, hotpotqa, musique). The evaluations use Claude3 Haiku as the downstream LLM.

## 5 ABLATION STUDY

In this section, we present the result of our ablation study for the proposed algorithm. We first show the impact of using another embedding model and ranking algorithm other than PageRank to construct a similarity matrix and the saliency of the phrases. Table 6 presents 4 scenarios in FRAPPE. The first row indicates the setup we have used throughout this paper, where the SBERT (all-MiniLM-L6-v2) model is used for the embedding model, and the PageRank algorithm is applied for ranking phrases. We have also experimented with another embedding model, GTE-large, a leading embedding model in the MTEB leaderboard, by fixing the ranking algorithm to the PageRank (the second row in the table). Different evaluation metrics on the MeetingBank dataset show that the algorithm is pretty stable w.r.t. to even a much smaller embedding model such as SBERT. Moreover, if we change the ranking algorithm to PacSUM (rows 3 and 4 in the table), which essentially uses an asymmetric centrality score (Equation 1) for ranking the phrases with both SBERT and GTE-large embedding models, we see almost the same results as the first row. All these suggest that using SBERT with the PageRank algorithm is a reasonable choice for FRAPPE to run our experiments in this paper.

| Methods | Scores for GPT3.5-Turbo | | | | |
|---|---|---|---|---|---|
| | ROUGE-1 | ROUGE-2 | ROUGE-L | BERTScore | METEOR |
| SBERT - PageRank | 0.2710 | 0.1112 | 0.1980 | 0.8479 | 0.2708 |
| GTE-large - PageRank | 0.2686 | 0.1061 | 0.1946 | 0.8474 | 0.2668 |
| SBERT - PACSUM | 0.2644 | 0.1109 | 0.1916 | 0.8462 | 0.2672 |
| GTE-large - PACSUM | 0.2827 | 0.1181 | 0.2050 | 0.8508 | 0.2851 |

Table 6: Ablation study on two different embedding models and two different ranking algorithms for the MeetingBank summarization task with a 2.5x compression ratio.

## 6 TOXICITY

To assess the impact of compression on toxicity we tested our model on the Toxigen dataset from Microsoft (Hartvigsen et al., 2022), which includes 250,000 samples with implicitly toxic and benign sentences about 13 minority groups. We used the popular detoxify (Hanu & Unitary team, 2020) to evaluate toxicity, employing both its "original" and "unbiased" versions. The detoxify models returns a probability of toxicity and classifications such as "severely toxic (sev toxic)", "obscene", "threat", "insult" and "identity attack (Id attack)". After establishing a baseline toxicity for each sample, we ran the samples through Frappe to compress the prompts and then re-evaluated their toxicity. In all categories, toxicity was significantly reduced, confirming that toxic language does not usually contribute anything meaningful to the conversations and can be effectively removed through compression. Frappe does not eliminate all toxic content, as the dataset contains highly concentrated toxic information and Frappe is not specifically trained for detoxification. However, across the board toxicity scores decreased, suggesting that the more inflammatory remarks were found to not be central to the underlying position. We aim to further study toxicity aware compression as if we explicitly do not want certain content in the outputs it may promise to be a prime compression candidate. Furthermore, we study the effects of of compression on toxicity and find that Frappe reduces toxicity by 50%. We also find that this is not the case with other compression algorithms, as some methods "distil" toxicity equating to a net overall increase in toxicity scores in their compressed outputs.

| Detoxify | Compression | Toxicity | Sev toxic | Obscene | Threat | Insult | Id Attack |
|---|---|---|---|---|---|---|---|
| Original | Uncompressed | 0.2840 | 0.0064 | 0.0431 | 0.0049 | 0.0746 | 0.1580 |
| | FRAPPE | 0.1679 | 0.0037 | 0.0238 | 0.0025 | 0.0381 | 0.0811 |
| | Lingua2-s | 0.4112 | 0.0165 | 0.0902 | 0.0076 | 0.1293 | 0.2150 |
| Unbiased | Uncompressed | 0.3184 | 0.0017 | 0.0108 | 0.0082 | 0.1645 | 0.2649 |
| | FRAPPE | 0.1712 | 0.0005 | 0.0050 | 0.0018 | 0.0776 | 0.1571 |
| | Lingua2-s | 0.3875 | 0.0034 | 0.0212 | 0.0058 | 0.1818 | 0.3496 |

Table 7: Detoxify toxicity scores using the "original" and "unbiased" models on the ToxiGen dataset with and without FRAPPE Compression.

## 7 CONCLUSION

Inspired by RAG, we proposed a simple yet efficient prompt compression method. Our approach, called FRAPPE, is task-agnostic and doesn't rely on any LLMs to generate data and compute conditional probabilities. In particular, FRAPPE first chunks the input prompt into phrases and removes some common uninformative phrases called "redundancies". Next, phrases closely aligned with these redundancies are also pruned from the prompt. This is done by measuring the similarity of their embedding vectors and removing those with a similarity above a threshold. Finally, using a graph-based ranking algorithm, the importance of the remaining phrases is computed, and the top ones are selected as the compressed input prompt. Comprehensive experiments show that the FRAPPE is up to 4 times faster than SOTA compression methods and often yields higher performance in a variety of downstream tasks.

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

# A    APPENDIX

This section includes the details of the experiments and some of the additional results, the effect of pruning articles and punctuation, more analysis of compression algorithms' latency, the trade-off between performance and compression rate in Frappe, and some representative samples of compressed input data.

## A.1    ARCHITECTURE DIAGRAM

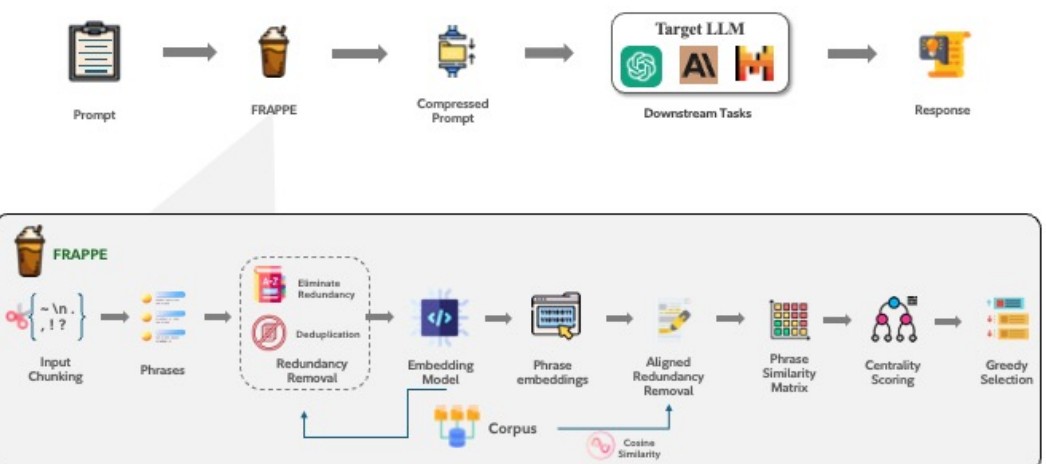

Figure 4: A high-level overview of our multi-step and modular approach.

## A.2    DETAILS OF EXPERIMENTS

In all the experiments, if the input's length exceeded the LLM's context length for uncompressed and compressed scenarios, we simply cut it off to fit it in the context window. We afforded the LLM a maximum of 400 tokens for its abstract.

**Summarization.** Given a text, the task is to generate a summary that captures the text's main points. The input prompt includes a short instruction for summarization, and the context and query are typically the input text. We use different evaluation metrics, including BLEU (Papineni et al., 2002), Rouge-1/2/L (Lin, 2004), METEOR (Banerjee & Lavie, 2005), and BertScore-F1 (Zhang et al., 2019).

**Single/Multi-Document QA.** The goal of this task is to evaluate the understanding of a model given a set of questions and the provided information in the document to generate the desired answers. The input prompt includes an instruction, a single or multi-document as the context, and a set of questions as the query. We use the F1 and Rouge-L scores provided in the LongBench benchmark (Bai et al., 2023) benchmark.

**Conversation.** Here, we want to generate an answer to a query given a previous conversation history. The above summarization scores are used to evaluate this task.

**In-context Reasoning.** This task measures the reasoning power of a model on complex tasks, including math and science. For this task, we use the Exact Match (EM) score, defined as the string's exact match between the prediction and the reference texts, to evaluate the compression method.

**Few-Shot, Synthetic Tasks, and Code Completion.** These tasks are from the LongBench benchmark. The few-shot is to accomplish a task (e.g., answering a question) given a few examples. F1 and Rouge- scores L are used as the evaluation metrics. The Synthetic tasks are similar to QA, e.g., What is the total number of different paragraphs in a given essay? Accuracy is the standard metric for this task. The code completion task predicts the next line of code given one or several pieces of code. The evaluation metric is Edit Similarity (similarity of two strings based on the number of insertions, deletions, and substitutions).

**Datasets and Benchmarks** For summarization, Other public datasets we have used for summarization include MeetingBank (Hu et al., 2023) (including 1,366 meetings transcripts from six cities or municipalities) and Arxiv preprint repository spanning topics such as Physics, Astrophysics, Biology, and Chemistry (Cohan et al., 2018). We have used 500 example articles from the test set of the `ccdv/arxiv-summarization`[9]. The LongBench (Bai et al., 2023) is a benchmark for multi-task assessment of long context understanding capabilities of LLMs. It comprises tasks including summarization, single/multi-document QA, few-shot learning, synthetic tasks, and code completion. Moreover, we have used the ZeroScrolls dataset (Shaham et al., 2023), consisting of ten different datasets and associated tasks. The datasets vary significantly in length, and the tasks span text summarization, query-based summarization, question answering, multiple-choice question answering, and aggregation. Each dataset and task pair has an associated metric, and to create a single "Zero-SCROLLS Score" for a particular algorithm, one takes the average of the results of the ten metrics across the ten datasets. We use GSM8K (Cobbe et al., 2021) and BBH (Suzgun et al., 2022) datasets for the reasoning task. GSM8k (Grade School Math 8K) is a dataset of 8.5K high-quality linguistically diverse grade school math word problems. BBH (Big Bench Hard) consists of a suite of 27 language and symbolic reasoning tasks spanning more than 6,500 problems, designed to evaluate chain-of-thought prompting. For our experiments, we have focused on 16 tasks as follows: temporal sequences, disambiguation qa, date understanding, tracking three shuffled objects, penguins in a table, geometric shapes, ruin names, tracking seven shuffled objects, tracking five shuffled objects, logical deduction for three objects, hyperbaton, logical deduction for five objects, logical deduction for seven objects, movie recommendation, salient translation error detection, reasoning about colored objects. Finally, We use the ShareGPT (sha, 2023) dataset for the conversation task.

### A.3 EXPERIMENTS WITH CLAUDE-3 HAIKU AND MISTRAL

Tables 8 and 9 demonstrate the performance of all compression algorithms on the Arxiv articles , MeetingBank transcripts, and ShareGPT conversations using Claude-3 Haiku and Mistral-8x7B model as the target LLMs, respectively. As we can see, FRAPPE achieves about the same or even better performance compared with other compression methods, and very close to the uncompressed one, while it is Faster than all other algorithms.

### A.4 EXPERIMENTS ON LONGBENCH

We ran on the LongBench test set, restricting use to only the English language tasks and the code tasks. As before, we prune 70% of tokens from each of the inputs. Since the LongBench task uses longer context windows, we elected to run this task through only the Claude-3 Haiku model due to the limit of the context window of the GPT-3.5 Turbo model. Table 10 illustrates LongBench scores for 4 different tasks of multi-doc-QA, summarization, FewShot, and code completion. As we can see our method remains competitive in all categories and is best for summarization-related tasks. This dataset shows that FRAPPE generalizes well to new domains as well as remains competitive with longer context windows. In addition, Table 11 shows the compression time on this dataset. As other experiments, FRAPPE has much faster running time compared to other algorithms.

---

[9]https://huggingface.co/datasets/ccdv/arxiv-summarization

| Methods | Rouge-1 | Rouge-2 | Rouge-L | BERTScore | METEOR | BLEU | Time(s) |
|---|---|---|---|---|---|---|---|
| | | | **Arxiv Articles** | | | | |
| Uncomp | 0.4178 | 0.1554 | 0.2254 | 0.8439 | 0.3302 | 0.0433 | – |
| SC | 0.3738 | 0.1159 | 0.2025 | 0.8355 | 0.2861 | 0.0213 | 3.9 |
| Lingua2-S | 0.3704 | 0.1185 | 0.2011 | 0.8353 | 0.2884 | 0.0219 | 0.23 |
| Lingua2-L | 0.3769 | 0.1227 | 0.2042 | 0.8372 | 0.297 | 0.0246 | 0.58 |
| **(FRAPPE)** | 0.3902 | 0.1355 | 0.2144 | 0.8392 | 0.3037 | 0.0362 | 0.14 |
| | | | **MeetingBank Transcripts** | | | | |
| Uncomp | 0.2462 | 0.1195 | 0.18 | 0.8428 | 0.3153 | 0.0426 | - |
| SC | 0.2077 | 0.0588 | 0.1367 | 0.8295 | 0.2435 | 0.0098 | 2.76 |
| Lingua2-S | 0.2382 | 0.0908 | 0.162 | 0.8397 | 0.2918 | 0.0237 | 0.11 |
| Lingua2-L | 0.2358 | 0.0882 | 0.1584 | 0.8392 | 0.2893 | 0.0219 | 0.23 |
| **(FRAPPE)** | 0.2204 | 0.0865 | 0.1569 | 0.8347 | 0.2662 | 0.0259 | 0.06 |
| | | | **ShareGPT Conversations** | | | | |
| SC | 0.4558 | 0.2306 | 0.3458 | 0.8806 | 0.3683 | 0.1531 | 0.71 |
| Lingua2-S | 0.4434 | 0.2005 | 0.3179 | 0.8777 | 0.3543 | 0.1185 | 0.06 |
| Lingua2-L | 0.4428 | 0.2015 | 0.3164 | 0.8780 | 0.3553 | 0.1132 | 0.14 |
| **(FRAPPE)** | 0.4374 | 0.2148 | 0.3179 | 0.8765 | 0.3500 | 0.1386 | 0.05 |

Table 8: Comparing FRAPPE with the SOTA methods on Arxiv articles, Meetingbank, ShareGPT using Claude 3 Haiku model as the target LLM and the compression rate of 0.7.

| Methods | Rouge-1 | Rouge-2 | Rouge-L | BERTScore | METEOR | BLEU | Time(s) |
|---|---|---|---|---|---|---|---|
| | | | **Arxiv Articles** | | | | |
| Uncomp | 0.4157 | 0.1548 | 0.2306 | 0.8408 | 0.3084 | 0.0483 | – |
| SC | 0.3638 | 0.1066 | 0.1958 | 0.8322 | 0.2666 | 0.0194 | 3.9 |
| Lingua2-S | 0.3604 | 0.1065 | 0.1962 | 0.8322 | 0.2725 | 0.0196 | 0.23 |
| Lingua2-L | 0.3595 | 0.1068 | 0.1966 | 0.8326 | 0.2734 | 0.0215 | 0.58 |
| **(FRAPPE)** | 0.3917 | 0.1299 | 0.2133 | 0.837 | 0.2896 | 0.0347 | 0.14 |
| | | | **MeetingBank Transcripts** | | | | |
| Uncomp | 0.2823 | 0.1386 | 0.2068 | 0.8510 | 0.3257 | 0.0514 | - |
| SC | 0.2292 | 0.0636 | 0.1468 | 0.8348 | 0.2482 | 0.0101 | 2.76 |
| Lingua2-S | 0.2374 | 0.0859 | 0.1586 | 0.8394 | 0.2804 | 0.0203 | 0.11 |
| Lingua2-L | 0.2362 | 0.0836 | 0.1557 | 0.8393 | 0.2782 | 0.0201 | 0.23 |
| **(FRAPPE)** | 0.2413 | 0.0914 | 0.1707 | 0.8413 | 0.2662 | 0.0278 | 0.06 |
| | | | **ShareGPT Conversations** | | | | |
| SC | 0.5364 | 0.1965 | 0.2924 | 0.8382 | 0.3774 | 0.0805 | 0.71 |
| Lingua2-S | 0.4906 | 0.1848 | 0.2763 | 0.8474 | 0.3412 | 0.0698 | 0.06 |
| Lingua2-L | 0.5834 | 0.2510 | 0.3928 | 0.8603 | 0.4253 | 0.0871 | 0.14 |
| **(FRAPPE)** | 0.4921 | 0.2345 | 0.3318 | 0.8791 | 0.3413 | 0.1419 | 0.05 |

Table 9: Comparing FRAPPE with the SOTA methods on Arxiv articles, Meetingbank, ShareGPT using Mistral-8x7B model as the target LLM and the compression rate of 0.7.

## A.5 PRUNING ARTICLES AND PUNCTUATION

In this section, we study the effect of forcing to prune articles (a/an/the) and punctuation in our compression algorithm. We have observed that removing articles and punctuation has little to no effect on the performance of our method, making it a good task-agnostic candidate. While this study has been done on summarization tasks, and probably for that task involve punctuation, such as math or logical operations, it might be hurtful to remove them; however, our experiments on GSM8K datasets in Section 4.1 have verified a little effect on the performance by removing punctuation on this dataset. Table 12 presents two sets of experiments for all 3 target LLMs: Rows starting with "Preserve" indicate preserving the articles and punctuation, while those starting with "Prune" indi-

| Methods | MultiDoc-QA | Summarization | FewShot | Code | Avg |
|---|---|---|---|---|---|
| **LongBench with Claude-3 Haiku** | | | | | |
| Uncomp | 40.8267 | 27.3600 | 26.0650 | 9.0600 | 25.8279 |
| SC | 34.2767 | 24.1833 | 33.0967 | 18.4800 | 27.5092 |
| Lingua2-S | 39.0567 | 24.7400 | 34.2433 | 16.9200 | 28.7400 |
| Lingua2-L | 38.7367 | 25.0933 | 33.2800 | 16.6250 | 28.4337 |
| **(FRAPPE)** | 37.5833 | 25.7567 | 31.4200 | 16.9100 | 27.9175 |
| **LongBench with GPT 3.5 Turbo** | | | | | |
| Uncomp | 32.0862 | 24.3825 | 29.0859 | 47.3140 | 33.2171 |
| Lingua2-S | 38.5577 | 22.7671 | 26.4245 | 36.1130 | 30.9656 |
| Lingua2-L | 38.2821 | 23.0461 | 26.0164 | 36.1700 | 30.8786 |
| **(FRAPPE)** | 35.4876 | 24.1804 | 26.5750 | 36.6290 | 30.7180 |

Table 10: Comparing our proposed approach with the SOTA methods for LongBench evaluation tasks using Claude-3 Haiku and GPT-3.5 Turbo Haiku models as the target LLMs and the compression rate of 0.7.

| Methods | **LongBench Compression Time(s)** | | | | Avg |
|---|---|---|---|---|---|
| | MultiDoc-QA | Summ | FewShot | Code | |
| SC | 7.03 | 4.21 | 5.54 | 3.86 | 5.16 |
| Lingua2-L | 0.75 | 0.60 | 0.66 | 0.51 | 0.63 |
| Lingua2-S | 0.37 | 0.32 | 0.36 | 0.25 | 0.32 |
| **(FRAPPE)** | 0.18 | 0.13 | 0.18 | 0.10 | 0.15 |

Table 11: Running time comparison of our proposed approach with the SOTA methods for Long-Bench evaluation tasks and the compression rate of 0.7.

cate results by forcing to remove them. As we can see, the summarization performance has stayed almost the same for both cases across all models, suggesting that we can easily let the algorithm remove articles and punctuation and simply gain a 1.2x compression ratio.

| **Model** | Rouge-1 | Rouge-2 | Rouge-L | BERTScore | METEOR | $1/\tau$ |
|---|---|---|---|---|---|---|
| Preserve-GPT-3.5 Turbo | 0.2830 | 0.1268 | 0.2100 | 0.8510 | 0.2935 | 1x |
| Prune-GPT-3.5 Turbo | 0.2821 | 0.1273 | 0.2090 | 0.8511 | 0.2980 | 1.2x |
| Preserve-Mistral-8x7B | 0.2823 | 0.1386 | 0.2070 | 0.8510 | 0.3257 | 1x |
| Prune-Mistral-8x7B | 0.2719 | 0.1296 | 0.1979 | 0.8487 | 0.3191 | 1.2x |
| Preserve-Claude3-Haiku | 0.2462 | 0.1195 | 0.1800 | 0.8428 | 0.3153 | 1 |
| Prune-Claude3-Haiku | 0.2450 | 0.1184 | 0.1787 | 0.8433 | 0.3147 | 1.2x |

Table 12: The effect of preserving and pruning articles and punctuation using different metrics for different models and configurations.

## A.6 TRADE-OFF BETWEEN PERFORMANCE AND COMPRESSION RATE

To start with the experiments, we run a representative experiment to monitor the performance of our approach versus different compression rates $(1 - \tau)$. This helps us understand

Here, we study the trade-off between Frappe's performance and compression rates using different combinations of embedding models and ranking algorithms. In particular, Figure 5 illustrates we monitor the BertScore between the summary of the compressed transcripts passed to the Claude-3 Haiku Model and the ground-truth of the MeetingBank dataset. As we can see, there is a slight change in the BertScore even with the compression rate as high as 90%, suggesting that FRAPPE can preserve the essential information in the input even with a very high compression rate.

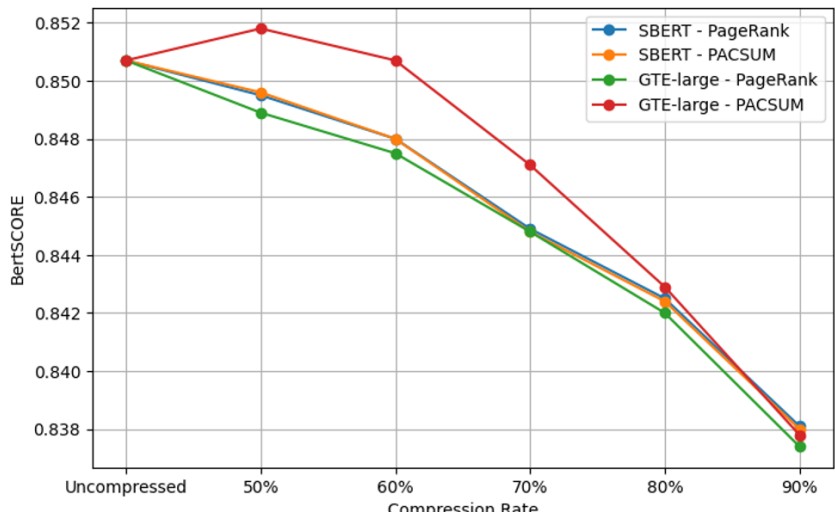

Figure 5: BertScore versus Compression Rate $(1 - \tau)$ on MeetingBank transcripts after passing the compressed transcripts to GPT-3.5-Turbo.

### A.7 REDUNDANCY GROUPS

Redundancy is a common occurrence in natural languages that tends to diminish the semantic significance of the data. Consequently, eliminating such redundancies can increase information density, thereby enhancing the overall quality and relevance of the input prompt. To address this issue, we have identified and established four categories of redundancy, as illustrated in Fig.2: Phatic Expressions, Filler Utterances, Connectives, and Stopwords.

**Phatic Expressions.** These comprise conversational phrases that primarily aim to foster or sustain social relationships rather than to relay specific information. We have initiated the process by asking GPT-4 for a comprehensive list of unique Phatic Expressions with the following specific prompts:

- Can you provide a comprehensive list of non-repetitive Phatic Expressions?
- Can you provide some more variations?
- Please provide a comprehensive list of Phatic expressions used for greeting in conversations.
- Please provide a comprehensive list of Phatic expressions used at the end of conversations.

To expand on this, we further prompted the model to provide additional variations and specific examples of Phatic expressions used in greetings and conversation closures. Some examples include "Hello", "Hi", "Hey", "how are you doing?", "Have a good one". After careful curation and verification, we compiled a list of approximately 90 Phatic expressions. To diversify this list, we have also sourced examples from online resources.

**Filler Utterances.** The second category involved prompting GPT-4 with the query, "Can you provide a comprehensive list of Filler Utterances in English?" This category aims to capture and address linguistic redundancies often found in casual conversations and discourse. We have compiled a list of approximately 30 words/phrases (both lower and upper case) like: "huh", "mmm", "whu", "uhm", "ah", "em", "umh", "eh", "um", "ha", "heh", "uh", "uhs", "wha", "mhm", "hum", "hmm", "humm", "oh", "uh-oh", "er" , "errr", "well", , "like", "actually", "basically", "seriously", "literally", "totally", "clearly", "you see", "you know", "I mean", "you know what I mean?", "at the end of the day", "believe me", "I guess", "I suppose", "or something", "so", "right".

**Connectives.** For this category, we have prompted GPT-4 with several queries aimed at generating a list of connective words, conjunctions, and transitional phrases in English. This exercise produced an extensive list of words, highlighting diverse groups such as Comparative (e.g., "similarly", "in the same way", "likewise"), Additive (e.g., "and", "also", "as well as", "moreover", "additionally"),

Contrastive (e.g., "but", "however", "on the other hand", "alternatively", "otherwise", "instead"), and others including Conditional, Summarize, Illustrative, and Time categories. Specific prompts used:

- Can you provide a comprehensive list of connective words in English?
- Can you provide a comprehensive list of conjunctions in English?
- Can you provide a comprehensive list of transitional phrases in English?

**Stopwords.** These refer to commonly used words that are frequently filtered out in natural language processing due to their minimal semantic content. In this study, we employed the NLTK stopword library(Bird et al., 2009).

### A.8 EXAMPLES OF COMPRESSED TEXT

In Fig. 6, we show a snippet of an uncompressed article from the Arxiv dataset and the text that results after compression with FRAPPE, aiming for a compression rate of 0.7. Similarly, In Fig. 7, we show a snippet of an uncompressed transcript from the Meetingbank dataset and the text that results after compression with FRAPPE, aiming for a compression rate of 0.6. Throughout the experiments presented in this paper, FRAPPE is not adjusted in any way when applied to different datasets.

---

**Snippet from original article (503 tokens):**

single - transverse spin asymmetries ( ssas ) play a fundamental role for our understanding of qcd in high - energy hadronic scattering . they may be obtained for reactions in , for example , lepton - proton or proton - proton scattering with one transversely polarized initial proton , by dividing the difference of the cross sections for the two settings of the transverse polarization by their sum . there have been extensive experimental investigations of such asymmetries @xcite . these have initiated much theoretical progress , in particular within the last few years . a particular focus has been on a class of single - spin observables that are characterized by a large momentum scale @xmath1 ( for example , the virtuality of the photon in deeply - inelastic scattering ( dis ) ) and by a much smaller , but also measured , transverse momentum @xmath2 . in such a `` two - scale '' situation , single - spin asymmetries may arise at leading power , that is , not suppressed by an inverse power of @xmath1 . for some of these cases , factorization theorems have been established @xcite that allow to write the spin - dependent cross sections in terms of parton distribution functions and/or fragmentation functions , perturbative hard - scattering functions , and so - called soft factors . a crucial feature is that the distribution functions and the soft factor in this factorization are not integrated over the transverse momenta of partons , because these in fact generate the observed transverse momentum @xmath2 . among other things , the observables may therefore provide valuable insights into the dependence of parton distributions in nucleons on transverse momentum . this becomes particularly interesting when the nucleon is transversely polarized , because there may be correlations between the nucleon spin vector , its momentum , and the parton s transverse momentum . one particular correlation , known as the `` sivers effect '' and described by so - called `` sivers functions '' @xcite , is now widely believed to be involved in a variety of observed hadronic single - spin phenomena . closer theoretical studies have revealed that the sivers effect plays an important role in qcd , beyond giving rise to phenomenological functions to be used in the description of single - spin asymmetries . a particularly interesting feature is that the sivers effect is not universal in the usual sense , that is , it is not represented by universal probability functions convoluted with partonic hard - scattering cross sections .

**Snippet after compression (146 tokens):**

single transverse spin asymmetries ssas play fundamental role for our understanding of qcd in high energy hadronic scattering particular focus has been on class of single spin observables are characterized by large momentum scale @xmath1 transverse momentum @xmath2 factorization theorems have been established @xcite allow to write spin dependent cross sections in terms of parton distribution functions and/or fragmentation functions because these generate observed transverse momentum @xmath2 observables may provide valuable insights into dependence of parton distributions in nucleons on transverse momentum this becomes interesting when nucleon is transversely polarized parton s transverse momentum it is not represented by universal probability functions convoluted partonic hard scattering cross sections

---

Figure 6: Comparison of text snippets from an uncompressed article from the Arxiv dataset with the compressed version of the snippet resulting from FRAPPE compression with a compression rate of 0.7. Tokens of the original text that have been pruned by the compression algorithm are highlighted in blue.

**Snippet from Original transcript (212 tokens):**

agenda item six council vote 119 348 related historic preservation imposing controls upon the halt hall, a landmark designated by the land risk preservation boards committee, recommends a bill passed. this rare picture. thank you very much. and hall hall is in the you district is 711 northeast 43rd. it was actually moved the building was built in 1928 and they moved it from where i-5 is now over to where it is located on 43rd street. this would be designated for the control features for the building exterior. the owner of the building was with us at our committee meeting and we would like to move forward with this landmark designation and move adoption of council bill 119348. any comments on this bill? please call the roll on the passage of the bill. gonzalez i. herbold hi. johnson i was i. o'brien i. want i. to make sure president herrell i adan favor an unopposed. bill passed and chair senate. please read the next agenda item. the short title.

**Snippet after Compression (74 tokens):**

agenda item six council vote 119 348 related historic preservation imposing controls upon halt hall recommends bill passed owner of building was us at our committee meeting we would to move forward this landmark designation move adoption of council bill 119348 any comments on this bill please call roll on passage of bill to make sure president herrell i adan favor unopposed bill passed chair senate

Figure 7: Comparison of text snippets from an uncompressed transcript from the Meetingbank dataset with the compressed version of the snippet resulting from FRAPPE compression with a compression rate of 0.6. Tokens of the original text that have been pruned by the compression algorithm are highlighted in blue.

