# OpenReview forum: "FRAPPE: Fast RAG-Inspired Prompt Evaporator"
_ICLR.cc/2025/Conference — Submitted to ICLR 2025_

### Official Review · Reviewer_xsWf · 2024-10-24

**Soundness:** 3
**Presentation:** 4
**Contribution:** 3
**Rating:** 5
**Confidence:** 3

**Summary:**

This paper proposes "Fast RAG Inspired Prompt Evaporator", or FRAPPE for short, which compresses the prompt to LLMs. It first processes the data to categorize and rank the phrases based on their informativeness. Only the phrases with the highest ranks will be selected as input for LLMs. FRAPPE achieves similar performance as the full-context baseline while greatly reducing the inference cost.

**Strengths:**

1. FRAPPE uses a suite of efficient and effective solutions to remove redundant phrases, including four categories: phatic expressions, filler utterances, connectives, and stop words. Each category is addressed with different methods, including the usage of neural-network-based similarity measurement.
2. The phrase ranking of FRAPPE deploys a graph-based ranking algorithm to identify the informative phrases.
3. FRAPPE can work in both task-agnostic and task-aware modes, giving more flexibility to users.
4. The performance of FRAPPE is superior to its baseline method LlmLingua.

**Weaknesses:**

1. One issue behind phrase-based compression is that it may break the meaning of a paragraph. The removal of some short phrases may completely change the meaning of a passage.
2. The compression method reduces the context length for LLM inference. However, its practical usage can be limited. On the one hand, the major overhead of LLMs, i.e. the inference cost for proprietary LLMs or the computation of local LLMs, is on generation instead of pre-filling. On the other hand, the compression rate of FRAPPE is low. A compression rate of 70% may not compensate for the performance loss or the extra step of the compression algorithm.
3. The compression rate is low. FRAPPE only removes phrases and is unable to merge phrases or shorten the text, e.g. making a shorter summary, thus its theoretical compression rate is bounded.
4. Measuring the latency of LLM APIs is less effective than directly measuring the runtime of local LLMs. The author should consider running inference with open-source LLMs if computation is allowed.

**Questions:**

1. Using the rule-based system to identify phrases can be fast but will result in long phrases. Have you ever considered using syntactic parsing to get a finer-grained unit?
2. Fig 4 in the appendix is a good explanation of the whole method but is rendered in low resolution. The authors should consider making it a vector graph.

---

> ### Author Response · Authors · 2024-11-15
>
> Thank you for taking the time to review our work and for highlighting both its strengths and areas for improvement. We're pleased that you recognized several key strengths of FRAPPE, including its efficient solution for removing redundant phrases, the flexibility offered by its task-agnostic and task-aware modes, and its superior performance compared to the baseline method, LLMLingua2. We greatly appreciate your insights and would like to address the weaknesses and questions you mentioned:
>
> [W1]: To address this issue, we’ve prioritized removing only filler words and phrases that do not affect the overall message, or those with low centrality scores based on the entire passage. This method helps maintain the core message of the text. Our experiments indicate that the accuracy of downstream tasks remains largely unaffected, and we have observed that large language models continue to provide accurate answers, suggesting no significant semantic loss.
>
> [W2]: While it's true that generation costs are a major factor in LLM inference, reducing input length can also offer cost benefits. In most scenarios where long inputs result in relatively short outputs, such as summarization or QA, compression becomes quite advantageous. For instance, in the MeetingBank dataset, where transcripts average 1-2K tokens and summaries are only 100-200 tokens, a 70% compression rate allows us to reduce input tokens to under 500, maintaining similar performance in downstream tasks. This can be particularly useful in retrieval-augmented generation (RAG) settings, where multiple documents can be compressed. Another setting is when a LLM is prompted with lengthy CoT in which the length of the input can dominate the LLM inference cost. All these common examples/applications of LLMs show the practical benefits of compression.
>
> [W3]: It's important to clarify that our method achieves a 70% compression rate, meaning we eliminate 70% of the tokens, not retain them (retaining only 30% of the original text). This detail is explained in Section 3: Proposed Architecture (lines 146-149), where we define the target compression rate as the fraction of pruned tokens, $\bar{\tau} = 1- \tau$ ($\tau$ denotes the fraction of remaining tokens). We will make this clearer in the paper to avoid any misunderstanding. While it's true that FRAPPE focuses on removing phrases rather than merging or shortening them, this approach provides the advantage of maintaining the integrity of the text, which can be crucial for preserving meaning and context. Additionally, our method offers flexibility by allowing different compression rates, potentially compressing content to a single phrase, depending on the task's requirements. This adaptability ensures that FRAPPE can be tailored to meet various needs, balancing compression rate with contextual integrity.
>
> [W4]: We acknowledge that directly measuring the runtime of local LLMs is more effective than measuring LLM API latency and would likely result in significant reductions in inference time. However, due to resource constraints by our institution on access to large GPUs, we were unable to conduct these tests.
>
> [Q1]: We chose a rule-based approach for phrase identification because it was intuitive and aligned with the design features of our algorithm. FRAPPE is specifically designed for standard English that includes punctuation. While syntactic parsing could provide more fine-grained units, it might also increase memory usage and latency. We prioritized maintaining the efficiency of the pipeline, but exploring syntactic parsing is definitely worth considering for future improvements.
>
> [Q2]: We appreciate your feedback regarding Figure 4 in the appendix. We will ensure that it is rendered as a high-resolution vector graphic to enhance the clarity and quality of the paper.

---

### Official Review · Reviewer_ZSpf · 2024-10-30

**Soundness:** 2
**Presentation:** 2
**Contribution:** 2
**Rating:** 3
**Confidence:** 3

**Summary:**

The paper presents a multi-stage, task-agnostic prompt compression methods to improve LLMs runtime latency and memory footprint. The proposed approach chunks context into text pieces and removed filler phrases and projects the text phrases into embedding space, ranked the phrases to keep only the most relevant ones using asymmetric centrality measure. The proposed approach could achieve a compression rate of 70%. Comparing with several other baseline methods, the approach achieves comparable or better accuracy performance and much lower latency. The authors also proposed context-aware setting to deal with high compression resulting loss of information related to query.

**Strengths:**

1. The proposed method is very empirical and is backed up by solid experiment results. The compression rate is high and performance on QA tasks looks very solid.
2. The authors take consideration of all kinds of issues in real application, e.g. they considered the possibility of high compression might lead to removing relevant info with respect to query, thus proposed context-aware solution.

**Weaknesses:**

1. Context compression/pruning comes with context info loss. It would be great if the authors could provide robustness analysis of the proposed approach. The concern is about the approach to remove filler words (such as articles) and chunking mentioned in the paper seems quite arbitrary. For instance, for questions like "If each child receives a cookie, how many cookies are needed for 10 children?", answering this kind of simple math question with removed articles might create ambiguous question, thus lead to incorrect answer.
2. The idea is very straightforward and intuitive, but it's very empirical as well. For instance, the idea with removing `Phatic Expressions` and `Contrastive (e.g. "but" as given in the paper)` is that it won't provide much help on answering the question. However, for some application like sentiment analysis, it would provide some value. Can we remove these info based on things such as mutual information etc? Or even simpler, it might be better just merge the two phases into one and using the proposed `asymmetric centrality` to remove the irrelevant info. It would be good to provide such experiments. Overall, the first step here gives the impression that we fallback to traditional NLP pipeline rather than leveraging LLMs' capability.

**Questions:**

1. If we remove articles in the context, would that be an issue for tasks such as GSM8K since it might be helpful to answer the questions? One example is `If each child receives a cookie, how many cookies are needed for 10 children?`
2. I assume when we do context pruning, we have to set a hyperparameter as the threshold, it would control the balance between info loss and compression rate. Would that be possible to show such tradeoff in the system? Latency could serve as an proxy, but would compression rate be more intuitive?

---

> ### Author Response · Authors · 2024-11-15
>
> Thank you for your thoughtful review and for highlighting both the strengths and areas for improvement in our work.
>
> [W1, Q1]: To ensure the robustness we tested the algorithm on 6 different datasets with multiple downstream tasks using 3 different LLMs. Like other papers, in our experiment's the instruction part of the input prompt is not compressed as it is typically short and is the source of guiding the target LLM. What matters for compression algorithms is to compress the provided context such as lengthy contexts on CoT, RAG pipeline, etc. For instance, on the GSM8K dataset, we only compress the context for math problems while leaving the instructions intact. Even with a 70% compression rate (70% of token are removed), we achieved 75.81% exact match accuracy on the GSM8K dataset,  compared to 80.06% with the full text. indicating that the removal of articles and similar elements does not significantly hinder performance in most cases. Unlike the contemporary methods, such as LLMlingua2, where mathematical operators like “+, -, *, /” and other key tokens are forced to be preserved, our approach does not impose such constraints. Instead, we leverage the inherent understanding of LLMs to process and interpret compressed text effectively. Moreover, we rely on the advanced capabilities of large language models (LLMs) to comprehend context, even when spelling and grammar are not perfect.
>
> [W2]: We agree that the removal of certain elements, such as phatic expressions or contrastive words like "but," can be context dependent and might not be ideal for tasks like sentiment analysis; however, our approach offers significant benefits in a task-agnostic context, which is the focus of our method. Our approach prioritizes simplicity and speed, aiming to develop an LLM-independent and task-agnostic pipeline. This aligns closely with the strengths of LLMs, allowing for broad applicability across various tasks without tailoring specifically to one.
> Moreover, relying on mutual information and other information theoretical measure has the issue that they need some model assumption for the underlying probability distribution. The goal here is to prevent using an underlying LLM (as opposed to other approaches) to estimate these measures as it can be biased and make the overall process slow and resource intense. Here, we established that using traditional NLP pipeline rather than leveraging LLMs' capability can indeed super useful both in terms of accuracy and speed/memory complexity.
> Additionally, we have run experiments utilizing asymmetric centrality (please see lines 482-483 and Table 6 in section 5) and we are planning on exploring how it can be further refined.
>
> [Q2]: Yes, we illustrate the trade-off between information loss and compression rate in line 918: Figure 5 in the appendix section, which shows the relationship between BERTScore and compression rate $\bar{\tau}$ on MeetingBank transcripts after passing the compressed transcripts to GPT-3.5-Turbo. We varied compression rates from 50% to 90% and observed a decrease in accuracy (as measured by BERTScore) as compression increased. We can additionally bring this to the main body as it provides insight into how different compression levels can impact the balance between retaining information and achieving higher compression.

---

### Official Review · Reviewer_PEcu · 2024-10-31

**Soundness:** 2
**Presentation:** 3
**Contribution:** 2
**Rating:** 3
**Confidence:** 4

**Summary:**

This paper proposes a prompt compression pipeline to remove redundant and unimportant content, reducing LLM latency and memory consumption. The pipeline includes three components responsible for text chunking, redundancy removal, and text compression. The authors tested this approach on tasks such as summarization, question answering, in-context reasoning, and code completion, demonstrating that the proposed FRAPPE model achieves improved latency performance.

**Strengths:**

1. The paper compares several of the latest approaches across different downstream NLP tasks, demonstrating latency advantages.
2. The collected expressions, utterances, and other resources from GPT-4 could benefit the research community if published.

**Weaknesses:**

1. The compression rate was fixed at 0.7 across all model comparisons, with no results reported for other compression rates. This makes it unclear whether the proposed approach consistently outperforms others at varying rates, which diminishes the contribution of this work.

2. The paper lacks a baseline comparison with simple sentence compression methods such as [1]. This LSTM-based approach is lightweight and may also offer speed advantages.

3. The toxicity study using this proposed method does not enhance the speed or performance of this work. Toxicity reduction relies mainly on the safety training strategy of the LLMs, ensuring that even if the input contains toxic content, the output remains free of it. Thus, I would not regard toxicity reducation as a plus of this work.

[1] Sentence Compression by Deletion with LSTMs. https://aclanthology.org/D15-1042/

**Questions:**

1. in some cases, I observed that your proposed compression method is even better than uncompression method, such as Arxiv Articles results in Table 1. I thought that compression means somehow some salient information is remove Could you provide an explanation?

---

> ### Author Response · Authors · 2024-11-15
>
> We appreciate your questions and hope this clarifies the strengths and limitations of our approach. Thank you for your thoughtful review.
>
> [W1]: Regarding the fixed compression rate of 0.7, we have provided an ablation study in Figure 5 (appendix), showing the relationship between BERTScore, a measure of semantic similarity, and the target compression rate on MeetingBank transcripts processed by GPT-3.5-Turbo.  This study provides some insights into the performance of our approach at different compression rates. We are preparing experiments with different compression rates to add to the paper and provide it here.
>
> [W2]: We have compared our algorithm with SOTA compression methods (Selective context and LLMlingua models). Moreover, in contrast to the LSTM-based approach in “Sentence Compression by Deletion with LSTMs”, FRAPPE uses cosine similarity and page ranking method, which makes it faster than many state-of-the-art compression algorithms. The LSTM-based method may not be directly applicable to the general-purpose prompt compression task that FRAPPE targets. Additionally, LSTM models require training, which can introduce challenges such as gradient vanishing/explosion, particularly with very long prompts (LSTM notoriously known slow for training long-context texts), which is the case for our experiments. In contrast and more importantly, FRAPPE is a training-free approach that does not rely on training for pruning. This makes it exceptionally memory efficient and fast (as reported in the paper), avoiding the computational overhead and potential issues associated with training neural networks.
>
> [W3]: One of the benefits of using prompt compression is that it can help reduce the toxicity and biasedness in the provided context in the input prompt. We’ve observed a reduction in toxicity when using our method compared to other models like LLMlingua2, which tend to condense toxic content. By reducing the toxicity in the input prompt, we potentially improve the likelihood of generating non-toxic outputs from the downstream LLM.
>
> [Q1]: Compressed text can sometimes outperform uncompressed text in scenarios where models have limited context lengths, such as GPT-3.5 Turbo with a 4k context limit. Compression can help highlight key information by removing redundancy or irrelevant content causing LLMs to hallucinate, especially when it is central to the task at hand, as shown in the "Lost in the Middle" paper by Liu et al., 2023.

---

> ### Author Response · Authors · 2024-11-18
>
> Thank you for your insightful feedback. As requested, we have generated results using a range of compression rates from 60% to 90% (where only 10% of the tokens remain after pruning) on the MeetingBank dataset using the GPT-35-turbo model and have included these results below. Our findings indicate that FRAPPE performs comparably to other state-of-the-art compression methods while offering significantly improved speed.
>
> ### Compression Rate: 60%
>
> | Methods   | rouge1 | rouge2 | rougeL | BERTSCore_F1 | METEOR | BLEU  | Time(s) |
> |-----------|--------|--------|--------|--------------|--------|-------|---------|
> | Uncomp    | 0.283  | 0.1268 | 0.21   | 0.851        | 0.2935 | 0.0466| -       |
> | SC        | 0.2631 | 0.1788 | 0.1787 | 0.8449       | 0.2539 | 0.0191| 2.76    |
> | Lingua2-S | 0.2788 | 0.1979 | 0.1977 | 0.8496       | 0.2905 | 0.034 | 0.11    |
> | Lingua2-L | 0.2764 | 0.1952 | 0.1954 | 0.8494       | 0.2884 | 0.0317| 0.23    |
> | FRAPPE    | 0.271  | 0.198  | 0.1977 | 0.8479       | 0.2708 | 0.0349| 0.06    |
>
> ### Compression Rate: 70%
>
> | Methods   | rouge1 | rouge2 | rougeL | BERTSCore_F1 | METEOR | BLEU  | Time(s) |
> |-----------|--------|--------|--------|--------------|--------|-------|---------|
> | Uncomp    | 0.283  | 0.1268 | 0.21   | 0.851        | 0.2935 | 0.0466| -       |
> | SC        | 0.2502 | 0.0693 | 0.1673 | 0.8417       | 0.2352 | 0.011 | 2.76    |
> | Lingua2-S | 0.2676 | 0.095  | 0.1841 | 0.8474       | 0.2754 | 0.024 | 0.11    |
> | Lingua2-L | 0.2673 | 0.0947 | 0.1838 | 0.8474       | 0.2749 | 0.0238| 0.23    |
> | FRAPPE    | 0.2632 | 0.1014 | 0.1902 | 0.8456       | 0.2605 | 0.0301| 0.06    |
>
> ### Compression Rate: 80%
>
> | Methods   | rouge1 | rouge2 | rougeL | BERTSCore_F1 | METEOR | BLEU  | Time(s) |
> |-----------|--------|--------|--------|--------------|--------|-------|---------|
> | Uncomp    | 0.283  | 0.1268 | 0.21   | 0.851        | 0.2935 | 0.0466| -       |
> | SC        | 0.2323 | 0.0506 | 0.1533 | 0.8377       | 0.205  | 0.0056| 2.76    |
> | Lingua2-S | 0.257  | 0.0818 | 0.1739 | 0.8441       | 0.2569 | 0.0181| 0.11    |
> | Lingua2-L | 0.2553 | 0.0791 | 0.172  | 0.8444       | 0.2548 | 0.0162| 0.23    |
> | FRAPPE    | 0.2425 | 0.0823 | 0.1737 | 0.8423       | 0.2252 | 0.021 | 0.06    |
>
> ### Compression Rate: 90%
>
> | Methods   | rouge1 | rouge2 | rougeL | BERTSCore_F1 | METEOR | BLEU  | Time(s) |
> |-----------|--------|--------|--------|--------------|--------|-------|---------|
> | Uncomp    | 0.283  | 0.1268 | 0.21   | 0.851        | 0.2935 | 0.0466| -       |
> | SC        | 0.2029 | 0.0276 | 0.1374 | 0.8323       | 0.1651 | 0.0009| 2.76    |
> | Lingua2-S | 0.2382 | 0.063  | 0.1605 | 0.8407       | 0.218  | 0.0109| 0.11    |
> | Lingua2-L | 0.2392 | 0.0598 | 0.1584 | 0.8406       | 0.2174 | 0.0085| 0.23    |
> | FRAPPE    | 0.2141 | 0.0569 | 0.1522 | 0.8377       | 0.1822 | 0.0118| 0.06    |

---

### Official Review · Reviewer_dEiH · 2024-11-04

**Soundness:** 2
**Presentation:** 2
**Contribution:** 2
**Rating:** 5
**Confidence:** 3

**Summary:**

This paper proposes a context compression method, aiming to improve compute and memory efficiency for LLMs. The proposed method -- FRAPPE -- chunks input text into phrases and removes two types of phrases (1) redundant phrases which are pre-defined and (2) non-salient phrases, which are determined by similarity with other phrases, calculated with a sentence embedding models. Experiments are conducted on multiple dataset and models, and compared against previously proposed context compression methods. The authors further proposed an extension to FRAPPE -- which takes similarity with the query into account and demonstrates improved performance.

**Strengths:**

* The paper proposes a task-agnostic method to perform context compression, which is an active research area to improve inference time efficiency of LLMs.
* The proposed method connects to previously proposed unsupervised summarization methods (e.g. PacSum) which is interesting.

**Weaknesses:**

The clarity of the paper, as well as experiment set-up could be improved. Please see questions below.

**Questions:**

**Experiments / ablation set-up**:
* How much does the two stages of filtering each contribute to the performance / efficiency trade-off? It would be helpful to conduct an ablation study on this. I am mostly wondering if it is possible to remove stage 1, or incorporate the constraint into the centrality calculation which could make the method cleaner.
* What is the uncompressed results for ShareGPT (for Table 1)?
* To contextualize the metrics / results, it might also be helpful to report a baseline which randomly decide whether to keep a phase or not (but retaining the same number of tokens).


**Clarification questions**:
* It is unclear to me how exactly does the context-aware FRAPPE method work. It'd be helpful to flash out the algorithm in equations / pseudo-code.

* I am finding the motivation of the detoxification experiments confusing -- is the idea that after compressing the context, the model is less likely to generate toxic content? It seems like toxicity is evaluated on the compressed context, instead of the generated text conditioned on the compressed context. Please clarify the hypothesis / goal for the detoxification experiments.

**Suggestions on related work**: There has been previously proposed methods which encode input context in chunks and use embedding models to derive similarity and decide which context to use ([0] and [1]). Although the methods are different from FRAPPE, they are definitely relevant.


[0] https://arxiv.org/abs/2310.04408
[1] https://arxiv.org/pdf/2310.03025

---

> ### Author Response · Authors · 2024-11-15
> **Addressing Reviewer's questions**
>
> Thank you for your insightful questions and suggestions. We appreciate the opportunity to clarify and expand on our work. We revisit the paper and include your suggestions/questions in the final draft to increase readability.
>
> [Q1]: We recognize the importance of understanding how each stage of our method impacts the overall performance and efficiency of our method. Removing the first filtering stage can skew centrality scores towards redundancy if such phrases dominate the document. To thoroughly investigate this and other stages of the algorithm, we are conducting an ablation study to evaluate the effects of each stage. We provide the experiments here and will add the results of this study in Section 6. Additionally, as a part of our Future work, we are exploring the potential to integrate redundancy scores directly into the centrality calculation as either a constraint or a parameter.
>
> [Q2]: ShareGPT (sha, 2023) dataset doesn’t have any ground-truth summary to compare with. This dataset includes the conversation transcripts from a human interaction with ChatGPT. Our goal here is to evaluate the compressed transcripts by asking “GPT-3.5 Turbo” to summarize them (please see line 311-313). Because ShareGPT dataset doesn’t provide any ground-truth summary, we’ve used generated summaries on uncompressed text as a proxy for ground truth (line 312).  This has allowed us to evaluate FRAPPE's efficiency against other state-of-the-art compression methods.
>
> [Q3]: Thanks for the thoughtful suggestion. Introducing a random pruning baseline is indeed a valuable addition for providing further context to our results. Our primary goal was to demonstrate FRAPPE's efficacy against state-of-the-art methods and full prompts as in the Selective-Context (Li et al., 2023b) (please refer to Table 2, page 6, in the paper: https://arxiv.org/pdf/2310.06201), a random pruning baseline was evaluated in the phrase level similar to ours. It has been shown that their method outperformed this baseline. We have demonstrated that FRAPPE outperforms the Selective Context algorithm, indicating that our method would also surpass the random pruning baseline. We will consider incorporating a random pruning baseline in our analyses to provide a more comprehensive comparison.
>
> [Q4]: We add a pseudo-code for the context-aware FRAPPE method. The main difference between this approach and FRAPPE is that it uses Personalized PageRank for the centrality score computation. It calculates the similarity between the query and each phrase and uses this as the input of the Personalized PageRank, forcing the underlying Markov chains to start a set of preferred nodes (phrases). This causes skewing the ranking towards those preferred nodes (emphasizing on the context for ranking the phrases).
>
> ### Step 1: Compute Embeddings
> phrase_embeddings = ComputePhraseEmbeddings(phrases)
> query_embeddings = ComputeQueryEmbeddings(query)
>
> ### Step 2: Compute Cosine Similarity Scores
> phrase_similarity_scores = ComputeCosineSimilarity(phrase_embeddings)
> query_similarity_scores = ComputeCosineSimilarity(phrase_embeddings, query_embeddings)
>
> ### Step 3: Normalize Cosine Similarity Scores and Prepare Preference Vector
> preference_vector = Normalized(query_similarity_scores)
>
> ### Step 4: Initialize Transition Matrix  & Rank Vector
> transition_matrix = NormalizeByRows(phrase_similarity_scores)
> rank_vector = InitializeUniformVector(num_phrases)
>
> ### Step 6: Iteratively Compute the Rank Vector
> For _ in range(max_iterations):
>     &nbsp;&nbsp;&nbsp;&nbsp;new_rank_vector = ComputeNewRankVector(preference_vector, transition_matrix, rank_vector, damping_factor)
>     &nbsp;&nbsp;&nbsp;&nbsp;# Check for Convergence
>     &nbsp;&nbsp;&nbsp;&nbsp;If Converged(new_rank_vector, rank_vector, tolerance):
>         &nbsp;&nbsp;&nbsp;&nbsp;&nbsp;&nbsp;&nbsp;&nbsp;rank_vector = new_rank_vector
>         &nbsp;&nbsp;&nbsp;&nbsp;Break
>     rank_vector = new_rank_vector
>
> ### Step 7: Return the Final Rank Vector
> Return rank_vector

---

> ### Author Response · Authors · 2024-11-15
> **Addressing Reviewer's questions[continued]**
>
> [Q5]: One of the most important implications of compression algorithms is that it can help reduce the toxicity and biasedness in the provided context in the input prompt.  In our experiments (section 6), the detoxification experiments aim to show that FRAPPE reduces toxic content compared to other state-of-the-art compression algorithms like LLMLingua2. Theoretically, less toxic compressed text should decrease the likelihood of generating toxic outputs in LLMs.  Specifically, we’ve used the ToxiGen dataset from Microsoft, which contains 250,000 samples with both toxic and benign sentences in about 13 minority groups. To evaluate toxicity, we used the Detoxify tool, which provides both a probability of toxicity and specific classifications such as "severely toxic," "obscene," "threat," "insult," and "identity attack." First, we established a baseline toxicity score for each sample without any compression. We then applied FRAPPE to compress the text prompts and re-evaluated their toxicity. Although more experiments need to analyze a compression algorithm’s impact on LLM-generated output, our findings (Table 7) show that FRAPPE significantly reduced toxicity across all categories compared to other compression SOTA methods.
>
> Thank you again for your valuable feedback and we will update the related work and provide citations to these papers.

---

### Official Review · Reviewer_iNob · 2024-11-04

**Soundness:** 2
**Presentation:** 2
**Contribution:** 2
**Rating:** 3
**Confidence:** 3

**Summary:**

The paper proposes a new approach to improve efficiency in retrieval augmented generation (RAG), by compressing its input context. The proposed method goes through multiple stages of compressing, removing phrases that belongs to pre-determined sets of unnecessary words (3.1, 3.2). Then, in the last step, they form a graph with phrases in the document, and use TextRank algorithm to identify salient phrases. The proposed, unsupervised approach shows up speed gain compared to the baselines. However, I have some issues with their experimental setups and writing.  Please see the weaknesses section.

**Strengths:**

The proposed method is simple and efficient.
The proposed approach is evaluated against many tasks and LLMs and show decent gains compared to the baselines.

**Weaknesses:**

The writing of the paper is not rigorous or clear.
* For example, in line 218, what is \bar{\tau_r}? Variables should be introduced more carefully. (I know \bar{\tau} is introduced in 147 but the subscript was not introduced).
* In the introduction, the discussion about toxicity comes out of the blue in the last bullet, very difficult to understand.

Questions / issues about the experimental setups:
* Why not include other baselines such as ICAE? The current choice of baselines should be justified more carefully (and described).
* Does this approach work with variable compression rate (seems most results are on 70% compression rate? - which, means you are removing only 30% of the tokens?)
* How is Time measured in the result tables? Why not report the time for uncompressed method?
* How much is getting reduced at each step? It'd be good to report them, to see which of the steps are actually impactful. We need ablation study on step 3.1, 3.2, 3.3.
* The toxicity section (Section 6) is hard to follow.  Also why selective-Context method not presented here as a baseline here?

**Questions:**

* It’d be helpful to explain *why* the memory requirement is big for the other methods (394-395).
* What was actual hyper parameter value that is chosen (lambda 1, lambda 2) for each experiment?
* Citation formats need a pass (e.g., line 133, “Pan…” should be \citet. Same in line 256).
* Line 92 “ATo” -> “To”
* Line 288-90, it’d be good to elaborate a bit on the choice of LLMs, whether they are open-sourced or API driven, etc.
* Second paragraph of related work would benefit from some cutting. The first half lack citations, thus making claims without supporting them (e.g., how does prompt compression leads to improved performances?)
* Missing related work:
    * RECOMP https://arxiv.org/abs/2310.04408

---

> ### Author Response · Authors · 2024-11-15
> **Addressing Reviewer's questions**
>
> Thank you for your thoughtful review and suggestions, helping us enhance the clarity of our paper.
>
> [W: Q1]: We chose baselines that align with the task-agnostic nature of our approach and if they don’t need any pretraining and/or fine-tuning. This choice is crucial, and it makes them directly comparable to ours. ICAE and approaches like Gisting (Mu et al., 2023) and AutoCompressor (Chevalier et al., 2023) require pre-training and/or (instruction) fine-tuning as they are based on soft prompt compression, involving some extra soft tokens/memory slot to be trained. As a result, these approaches potentially are task dependent. However, FRAPPE, LlmLingua2 models (Pan et al., 2024) and Selective-Context (Li et al., 2023b) methods are task agnostic, and they can be seamlessly integrated into LLM pipelines without additional preparation.
>
> [W: Q2] We acknowledge the potential for misunderstanding and will ensure that this is communicated more clearly in the paper.
> To clarify, when we refer to a 70% compression rate, it means that we eliminate 70% of the tokens, retaining only 30% of the original text. This detail is explained in Section 3: Proposed Architecture (lines 146-149), where we define the target compression rate as the fraction of pruned tokens, $\bar{\tau} = 1- \tau$ ($\tau$ denotes the fraction of remaining tokens).  $\tau$ with subscripts “c, r, e” are the corresponding compression rate at the end of each three stages of our method: $\tau_c$ is the compression rate by the end of INPUT Chunking stage. $\tau_c$ is the compression rate by the end of ALIGNED REDUNDANCY REMOVAL stage, and $\tau_e$ denotes the compression rate by the end of PHRASE RANKING AND SELECTION stage. Moreover, the relation between the target compression rate and these stages is given by line 228.  We add this clarification at the beginning of section 3.
> Furthermore, we have explored a range of compression rates, from 50% to 90%, to demonstrate the flexibility of our approach. This analysis is presented in Figure 5 of the appendix, where we show the relationship between BERTScore, a measure of semantic similarity, and the target compression rate $\bar{tau} on MeetingBank transcripts processed by GPT-3.5-Turbo. As expected, the plot indicates that as compression increases, there is a corresponding decrease in accuracy, highlighting the trade-off between retaining information and achieving higher compression. We try to bring this plot to the main part of the paper by cutting the second paragraph of related work as suggested.
>
> [W: Q3] In our tables, the "Time" column refers to the per-input compression time, measured in seconds when using a NVIDIA A100 GPU. This represents the average time taken by our compression algorithm to process each input across the entire dataset. Time analysis of the FRAPPE and other algorithms using a GPU device has been presented in Figure 3. Regarding the uncompressed method, as there is no compression step involved when using uncompressed text, there is no compression time to report for this method (the input is processed in its entirety without any preliminary compression phase).
>
> [W: Q4] Thanks for the suggestion. We indeed are preparing experiments to evaluate compression achieved in each of the stages (3.1, 3.2 and 3.3). We will report it here as soon as possible and make sure to add to the ablation study as well.

---

> > ### Comment · Reviewer_iNob · 2024-11-26
> >
> > [W: Q1] Thanks for the explanation of the baseline choice. This helps! These other line of work, which might be of different focus, have the similar goal and should be compared. What we should do is to interpret the results carefully, comparing how much models that do not need additional preparation can improve vs. models that need additional preparation. In other words, you do not have to outperform these other baselines that I suggested, but they should be provided to contextualize your results.
> >
> > [Q2-4], it would be good to address this in the updated draft.

---

> ### Author Response · Authors · 2024-11-15
> **Addressing Reviewer's questions[continued]**
>
> [W: Q5] We will re-write the Toxicity section for better readability and clarification. In our study, we’ve explored the impact of our compression method, FRAPPE, on the toxicity of text. We used the ToxiGen dataset from Microsoft, which contains 250,000 samples with both toxic and benign sentences in about 13 minority groups. To evaluate toxicity, we used the Detoxify tool, which provides both a probability of toxicity and specific classifications such as "severely toxic," "obscene," "threat," "insult," and "identity attack." First, we established a baseline toxicity score for each sample without any compression. We then applied FRAPPE to compress the text prompts and re-evaluated their toxicity. Our findings show (Table 7) that FRAPPE significantly reduced toxicity across all categories. This suggests that toxic language often does not contribute meaningful content to the context of the input prompt and can be effectively minimized through compression. Unlike the contemporary compression methods that may increase toxicity by condensing toxic elements and emphasizing their importance, FRAPPE reduced the overall toxicity score, indicating that inflammatory remarks are not central to the core message. Finally, we did not include the Selective-Context method because it is slow and resource-intensive and slow as shown in all experiments. Furthermore, LLM-Lingua2 is currently the state-of-the-art and fastest model available apart from FRAPPE. However, we recognize the importance of including Selective-Context for a comprehensive comparison and plan to run it and share the results as soon as possible. Overall, FRAPPE reduced toxicity by about 50%, making it a promising approach for handling toxic content in text, while other methods may inadvertently increase toxicity by compressing toxic importance.
>
> [Q1] The high memory requirement of the other methods primarily stems from their use of large neural networks to evaluate the relevance of each token. For instance, LlmLingua2 is available in two sizes: a large model, FacebookAI/xlm-roberta-large, and a smaller model, google-bert/bert-base-multilingual-cased. Similarly, the selective context method utilizes models like GPT-2 or LLaMA2, which, although smaller than some of the largest models available, still require significant memory resources due to their architecture. In contrast, our approach, FRAPPE, uses a very small sentence transformer model, specifically the sentence-transformers/all-MiniLM-L6-v2. This model is lightweight and efficient, allowing us to achieve comparable accuracy with significantly lower memory requirements. By not relying on large models for determining phrase relevance, FRAPPE offers a more memory-efficient alternative without sacrificing performance.
>
> [Q2]: For the experiments involving the PacSum algorithm, we used the hyperparameter values ($\lambda_1 = -2$) and ($\lambda_2 = 1$). These values were chosen based on the optimal settings identified in the original PacSum paper, as indicated in Figure 1 of (Zheng & Lapata, 2019). We will include these details in the relevant sections of our paper to enhance transparency and assist others who may wish to replicate or build upon our work.
>
> [Q3 Q4 Q6 Q7]
> Thank you for your detailed feedback. We will address the formatting and content issues as follows: we'll correct citation formats, such as using \citet for the citation on line 133 and similar adjustments on line 256 and fix the typographical error on line 92 by changing "ATo" to "To." In the related work section, we'll include and cite the RECOMP paper (https://arxiv.org/abs/2310.04408) to provide a more comprehensive overview. These revisions will enhance the clarity and scholarly rigor of our paper.
>
> [Q5] In our experiments, we used closed-source LLMs, specifically GPT-3.5-turbo by OpenAI, Claude 3 by Anthropic, and Mistral-8x7B Instruct v0.1 by MistralAI. These models were selected because they demonstrated top-performing capabilities during our initial tests, particularly on low-end models. They offer state-of-the-art (SOTA) performance, which allowed us to thoroughly evaluate our algorithm across a wide range of datasets and downstream tasks. We will ensure that this information is clearly presented in the paper to provide readers with a comprehensive understanding of our choice of LLMs.

---

### Meta-Review · Area_Chair_uB6j · 2024-12-20

**Metareview:**

The paper proposes "Fast RAG Inspired Prompt Evaporator" (FRAPPE), a multi-stage, task-agnostic method to compress input prompts for large language models (LLMs). FRAPPE categorizes and ranks phrases based on informativeness, removing redundant or irrelevant content through several stages, including chunking, redundancy removal, and graph-based ranking using TextRank.


Weakness:
-  Baseline choices are insufficiently justified, excluding key methods like ICAE, even after rebuttal, Reviewer iNob mentioned that there are important baselines excluded.
- Poorly motivated toxicity reduction experiments with unclear relevance to compression (after rebuttal from Reviewer dEiH).

**Additional Comments On Reviewer Discussion:**

While FRAPPE demonstrates promising results in prompt compression for LLMs, the methodological and experimental weaknesses undermine its contributions. The lack of clarity, limited exploration of trade-offs, and insufficient baseline comparisons make it difficult to evaluate the approach's generalizability and practical utility. Addressing these issues could significantly strengthen the paper.

---

### Decision · Program_Chairs · 2025-01-22

Reject